# Recent Progress in Photon Upconverting Gels

**DOI:** 10.3390/gels5010018

**Published:** 2019-03-26

**Authors:** Pankaj Bharmoria, Nobuhiro Yanai, Nobuo Kimizuka

**Affiliations:** 1Department of Chemistry and Biochemistry, Graduate School of Engineering, Center for Molecular Systems (CMS), Kyushu University, 744 Moto-oka, Nishi-ku, Fukuoka 819-0395, Japan; pankajbharmoria@gmail.com; 2PRESTO, JST, Honcho 4-1-8, Kawaguchi, Saitama 332-0012, Japan

**Keywords:** photon upconversion, triplet-triplet annihilation, photoluminescence, oxygen blocking, self-assembly

## Abstract

Recent progress in the development of gels showing triplet-triplet annihilation based photon upconversion (TTA-UC) is reviewed. Among the two families of upconverting gels reported, those display TTA-UC based on molecular diffusion show performances comparable to those in solutions, and the TTA-UC therein are affected by dissolved molecular oxygen. Meanwhile, air-stable TTA-UC is achieved in organogels and hydrogels by suitably accumulating TTA-UC chromophores which are stabilized by hydrogen bonding networks of the gelators. The unique feature of the air-stable upconverting gels is that the self-assembled nanostructures are protected from molecular oxygen dissolved in the microscopically interconnected solution phase. The presence of the bicontinuous structures formed by the upconverting fibrous nanoassemblies and the solution phase is utilized to design photochemical reaction systems induced by TTA-UC. Future challenges include in vivo applications of hydrogels showing near infrared-to-visible TTA-UC.

## 1. Introduction

The excited triplet state plays pivotal roles in a variety of photofunctional systems. The recent development of triplet-triplet annihilation-based photon upconversion (TTA-UC) has added emergent importance to photochemistry [1,2,3,4,5,6,7,8,9,10,11]. The TTA-UC typically occurs in multi-chromophore systems, which is initiated by triplet energy transfer (TET) from a photo-excited donor (sensitizer) to an acceptor (emitter) via an electron exchange (Dexter energy transfer) mechanism. When two sensitized acceptor triplets diffuse and collide, annihilation occurs to give a higher energy singlet state (S_1_), from which the delayed anti-Stokes fluorescence is emitted. Due to its ability to function at subsolar irradiance and a wide spectral window, the TTA-UC has attracted broad interest for applications, including a solar cell, photocatalysis, bioimaging, photodynamic therapy, and drug delivery [12,13,14,15]. Meanwhile, the excited triplet states are susceptible to deactivation by molecular oxygen (^3^O_2_) [16,17], and their protection from oxygen is an outstanding issue in triplet-related photochemical processes [18]. This holds true for TTA-UC, and it has been investigated by completely deaerating the solutions or solid materials followed by the sealing procedures. However, the use of such deaerated samples limits the potential of TTA-UC to be maximiszd in many applications. Therefore, the development of air-tolerant TTA-UC systems is highly desired. Various design principles have been developed to solve this issue, such as polymer coated micro/nanoparticles [19,20,21,22,23,24], acceptor molecular assemblies [25,26,27,28,29], and supramolecular gels. In this article, we highlight gel materials that are designed to achieve TTA-UC under ambient conditions.

Gels are soft materials consisting of polymeric networks of covalent or non-covalent polymers that show unique rheological properties depending on the chemical structures of the networks and the solvents. There are mainly two approaches to achieve TTA-UC in gels; the molecular diffusion mechanism and energy migration mechanism. These photon upconverting gels are attractive due to their nonfluidity and good processability, which allow the preparation of transparent film-like materials. In addition, their internal solution phase has been recently utilized as the reaction media for triplet sensitized chemical reactions [30,31,32]. However, as the solution phase within gels contains dissolved molecular oxygen, it is a challenge to add air-stability to photon upconverting gels by avoiding the contact between excited triplet states and oxygen molecules. The molecular diffusion-based TTA-UC works very well in deaerated low-viscosity solution; however, it suffers from inevitable oxygen quenching in air-saturated gels. Meanwhile, the TTA-UC via triplet energy migration in densely organized chromophore assemblies can be efficient even in air-saturated gels. These chromophores are typically accumulated in the networks of hydrogen-bonded gelator assemblies, which are separated from the microscopically interconnected solution phase in gels. Therefore, the structure of molecular assemblies plays a crucial role in reducing the participation and diffusion of molecular oxygen inside gels that are required to avoid the triplet quenching.

In the present review article, we discuss the current status of upconverting gels with a particular focus on their air-stability. We review photon upconverting gels using various solvents, such as organic solvents (organogels), ionic liquids (ionogels), and water (hydrogels). It will contribute to the design of gelator structures to block molecular oxygen and to achieve efficient TTA-UC in air.

## 2. Upconverting Organogels

Organogels are generally composed of low-molecular-weight gelators or cross-linked polymers as the continuous phase and organic liquids as the dispersed phase. Reports on TTA-UC in organogels have been published by the research groups of Schmidt, Simon and Weder, and Yanai and Kimizuka. Schmidt and co-workers reported an upconverting organogel that contains palladium tetraphenylporphyrin (PdTPP) and 9,10-diphenylanthracene (DPA) as donor/acceptor chromophores, tetralin as organic solvent, and 1,3:2,4-bis(3,4-dimethylbenzylidene) sorbitol (DMDBS) as low-molecular-weight gelator (Figure 1) [33]. Interestingly, the organogel shows TTA-UC performance indistinguishable from the liquid sample. This is because all the TTA-UC events take place in the liquid cavities, which are larger than the average diffusion distance of triplet molecules (~300 nm) within a typical triplet lifetime of 100 μs. The organogel formed by DMDBS itself does not have the ability to block oxygen, and they demonstrated TTA-UC in air by immersing the organogel within an oxygen-scavenging water solution.

Weder and Simon et al. reported an upconverting organogel composed of *N*,*N*-dimethylformamide/dimethyl sulfoxide solution of Pd(II) mesoporphyrin IX (PdMesoIX) and DPA, and a three-dimensional polymer network of covalently cross-linked poly(vinyl alcohol) with hexamethylene diisocyanate (Figure 2a) [34]. The organogel showed a high TTA-UC efficiency (Φ_UC_´) of 14% under the deaerated condition, which is also ascribed to molecular diffusion of the donor and acceptor chromophores in the interconnected liquid compartments. The organogel was shown to exhibit the TTA-UC emission in air (Figure 2b), but its UC emission intensity was not stable under continuous excitation due to oxygen quenching. The addition of an oxygen scavenger in the liquid phase would improve the stability of the TTA-UC emission [35,36].

In contrast to the above TTA-UC gel systems in which the diffusion of excited triplet molecules in the institutinal liquid phase plays an essential part, Yanai and Kimizuka et al. developed air-stable supramolecular upconversion gels, which harness the triplet energy transfer and migration among donor and acceptor chromophores densely accumulated in the interior of fibrous gel nanofibers (Figure 3) [37]. The ternary supramolecular gels were obtained by mixing platinum octaethylporphyrin (PtOEP), DPA and gelator *N*,*N*′-bis(octadecyl)-l-boc-glutamic diamide (LBG) by heating in DMF and cooling to room temperature. Remarkably, the ternary PtOEP/DPA/LBG supramolecular gel showed stable TTA-UC emission in air, which is ascribed to the solvophobic self-assembly of dyes in the interior of gel nanofibers. Moreover, the observed low excitation intensity threshold, *I*th, of 1.48 mWcm^−2^ in the air-saturated gel reflects the effective triplet energy migration among the chromophores self-assembled in high density with proximity. The confinement of the donor and acceptor molecules inside the nanofibers and the presence of polymeric, multiple hydrogen bond networks among LBG molecules effectively shielded excited triplets from the molecular oxygen dissolved in the surrounding liquid phase. It is also noteworthy that when a non-polar solvent, such as carbon tetrachloride, was employed, the ternary gel did not show TTA-UC in air because the donor and acceptor stayed dissolved in the liquid phase and they underwent oxygen quenching. Thus, both the solvophobic interactions and the adaptive nature of the host LBG that maintained the integrity of the hydrogen-bonded nanofibers contributed to the observed TTA-UC performance in air. Supramolecular gel nanofibers provide a general methodology to realize TTA-UC in air from a wide combination of donor-acceptor pairs, allowing the near infrared (NIR)-to-yellow, red-to-cyan, green-to-blue, and blue-to-UV wavelength conversions. These features provide the solution for the outstanding issue of oxygen quenching in molecular diffusion-based upconverting gels [33,34].

New applications of the air-stable TTA-UC exerted by the self-assembled PtOEP/DPA/LBG organogels have been developed. Díaz and coworkers demonstrated the chemical transformation of aryl halides via single electron transfer from the singlet state generated by green-to-blue TTA-UC in PtOEP/DPA/LBG organogels under aerobic conditions (Figure 4) [30]. Interestingly, photoreduction of the aryl halide substrate occurred with a high mass balance and yield without the formation of byproducts. Furthermore, comparable results were obtained when the model reactions were performed under a nitrogen atmosphere, indicating that the efficient confinement effect of photoiunduced radical reactions in the self-assembled fibrous gel networks occurred. This report shows the beneficial use of both the upconverting self-assembled gel networks and the inner solution phase for the photochemical reaction, which provide a new perspective in synthetic chemistry.

Matrix-free upconverting organogels have also been reported by the Yanai and Kimizuka group [38]. Chiral nanoassemblies were self-assembled from lipophilically modified donor/acceptor pairs, which showed effective triplet energy migration in the gel nanofibers. Upon heating the organogels to 90 °C, the donor/acceptor co-assemblies underwent disassembly and resulted in the complete disappearance of the UC signal due to the rapid deactivation of the acceptor triplets through the conformational changes. Meanwhile, upon cooling back to 25 °C, the donor and acceptor molecules re-assembled to regenerate the TTA-UC emission (Figure 5). Thus, this system shows the switching of aggregation-induced photon upconversion, which is accompanied by the reversible sol–gel transition.

## 3. Upconverting Ionogels

Ionic liquids (ILs) have been popularized in chemistry and materials science because of nonvolatility, nonflammability, and the other unique properties, which are tunable depending on the combination of ionic constituents [39]. The physical gelatinization of ionic liquids by dissolved carbohydrates and self-assemblies was first reported by Kimizuka et al. in 2001 [40]. The name of ionogel refers to the solvents that are gelatinized, as is the case with the commonly used terms of organogel or hydrogel. The formation of ionogels provides a rational solution to circumvent the solvent-evaporation issue in common organogels.

Murakami and coworkers reported TTA-UC in ionogels [41]. The TTA-UC ionogel was prepared by dissolving the UC dye pair of palladium meso-tetraphenyl-tetrabenzoporphyrin (PdPh4TBP), perylene, and a polymeric salt gelator in IL; 1-butyl-2,3-dimethylimidazolium bis(trifluoromethylsulfonyl)amide ([C_4_dmim][NTf_2_]) (Figure 6). TTA-UC occurred based on the molecular diffusion in the ionogels under deaerated conditions. The ionogel showed excellent stability against flame thanks to the excellent nonflammability of ILs (Figure 6e).

## 4. Upconverting Hydrogels

Biological applications of upconverting gels require their function under aqueous environments. However, there are significant hurdles to achieve TTA-UC in aqueous environments due to the low chromophore solubility and the presence of dissolved molecular oxygen. In this direction, Wang and coworkers reported a TTA-UC microemulsion hydrogel, in which a toluene solution of chromophores (PdTTP and DPA) was emulsified in water using Tween 80 as a surfactant and mixed with sodium polyacrylate (PAAS) as an O_2_ blocking gelator (Figure 7) [42]. The hydrogel containing the oil-in-water microemulsion showed a TTA-UC emission in air at 60 °C. A weakened TTA-UC emission was observed at around room temperature due to the limited diffusion of chromophores in the nanoscale oil droplets. By utilizing this behavior, thermally-induced switching between TTA-UC emission and phosphorescence was reversibly achieved (Figure 7a).

TTA-UC hydrogels functioning in physiological conditions without the use of organic solvents would be suitable for biological applications. In a recent conceptual development, the Yanai and Kimizuka group prepared a series of upconverting hydrogels from a biopolymer-surfactant co-assembly without organic solvents (Figure 8) [43]. A series of differently charged biopolymers, like cationic gelatin, anionic sodium alginate, and neutral agarose, were co-assembled with non-ionic surfactants (Triton X-100, Tween 80, and Pluronic f127) as a host matrix for a upconversion (UC) dye pair of PtOEP and sodium 9,10-diphenylanthracene-2-sulfonate (DPAS). The biopolymers co-assembled with surfactants provided a thick O_2_ barrier to chromophores arranged inside the co-assembly for a long triplet lifetime of up to 4.9 ms and a high TTA-UC efficiency, Φ_UC_′, of 13.5% even in the air-saturated condition. A control experiment without biopolymers showed an unstable UC emission with a short triplet lifetime of 21 µs due to the oxygen quenching. The oxygen blocking ability of the coassembled hydrogel structure was further confirmed from a TTA-UC emission switching induced by the gel–sol transition. At 60 °C, 60% of the TTA-UC emission was quenched by oxygen in the disassembled sol phase. The biopolymer-surfactant-chromophore coassembly approach provides a simple and general methodology to achieve aqueous TTA-UC in air without time-consuming degassing processes.

## 5. Conclusions and Future Possibilities

The research on upconverting gels has established a lot of conceptual developments. The performance of molecular diffusion-based TTA-UC in gels was found to be comparable to that in solutions. The poor air stability of this type of upconverting gels would be improved by adding oxygen scavengers. Meanwhile, ordered molecular self-assembly offers another promising means to solve this issue. The design concept of upconverting gels with intrinsic air stability has been developed; chromophore assemblies installed in dense hydrogen bonding networks of the gelators. The generality of this concept has been confirmed in organogels and hydrogels, and thus the challenge of air-stable TTA-UC in the co-existence of the air-saturated, microscopically interconnected solution phase has been achieved in suitably designed supramolecular gels. The unique feature of air-stable upconverting gels has been demonstrated in the photochemical reaction system, which utilized the singlet state generated by TTA-UC. The emergence of air-stable TTA-UC hydrogels would foster biological applications. For their practical in vivo applications, future studies should be directed at limiting the use of harmful surfactants. Moreover, until now, a large portion of studies on upconverting gels has been limited to visible-to-visible TTA-UC, and this scope should be expanded to NIR-to-visible TTA-UC for in vivo applications and NIR-powered photochemistry.

## Figures and Tables

**Figure 1 gels-05-00018-f001:**
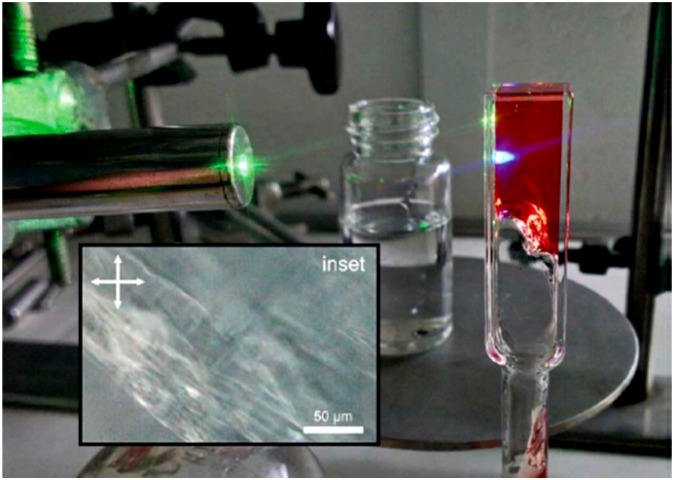
A photograph of 1,3:2,4-bis(3,4-dimethylbenzylidene) sorbitol (DMDBS)-palladium tetraphenylporphyrin (PdTPP)-9,10-diphenylanthracene (DPA) organogel in deaerated tetralin showing green-to-blue upconversion in an inverted cuvette. Adopted with permission from [33] Copyright 2015 Royal Society of Chemistry

**Figure 2 gels-05-00018-f002:**
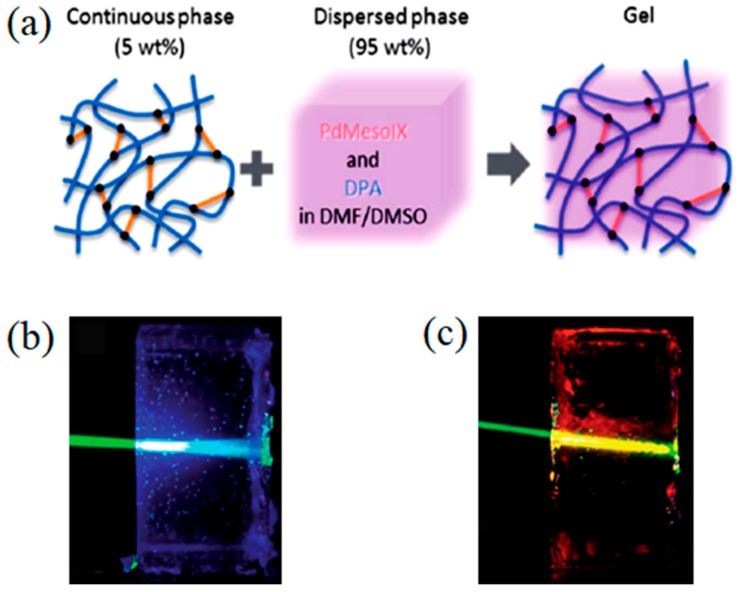
(**a**) Graphic representation of the structure of the upconverting organogel, (**b**) organogel containing both PdMesoIX and DPA pair showing green-to-blue triplet-triplet annihilation based photon upconversion (TTA-UC) emission upon 543 nm laser excitation under ambient conditions, (**c**) organogel containing only PdMesoIX showing no upconversion (UC) emission. Adopted with permission from [34]. Copyright 2015 Royal Society of Chemistry.

**Figure 3 gels-05-00018-f003:**
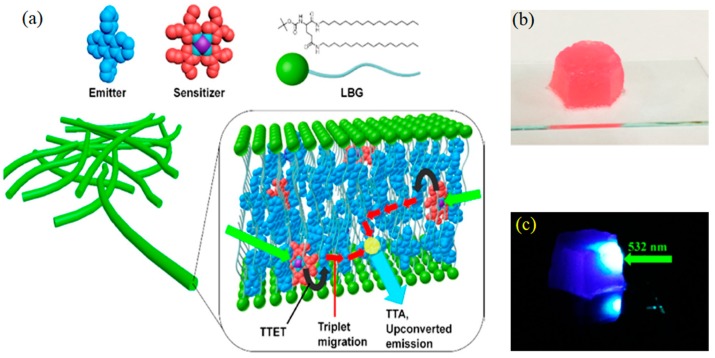
(**a**) Schematic representation of the unit structure of the upconversion gel system. Donor (red) and acceptor (blue) molecules are incorporated in the *N*,*N*′-bis(octadecyl)-l-boc-glutamic diamide (LBG) nanofibers as extended domains. The donor molecules are excited by long-wavelength light, followed by a sequence of triplet–triplet energy transfer (TTET), triplet energy migration (TEM), triplet-triplet annihilation (TTA), and delayed fluorescence from the upconverted singlet state of acceptor molecules. Pictures of the platinum octaethylporphyrin (PtOEP)/DPA/*N*,*N*′-bis(octadecyl)-l-boc-glutamic diamide (LBG) ternary gel shaped in a mold under (**b**) white light and the (**c**) 532 nm green laser in air. No filter was used to take the pictures. Adopted with permission from [37]. Copyright 2015 American Chemical society.

**Figure 4 gels-05-00018-f004:**
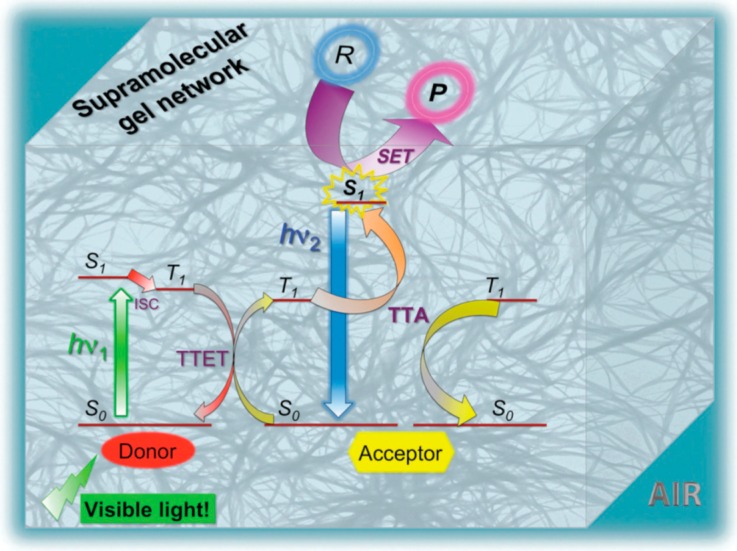
Schematic representation of chemical reactions powered by upconverted singlet energy. This event can be achieved using visible light at room temperature and in air when confined into a gel doped with a donor/acceptor pair. R = reactant; P = product. Adopted with permission from [30]. Copyright 2015 Royal Society of Chemistry.

**Figure 5 gels-05-00018-f005:**
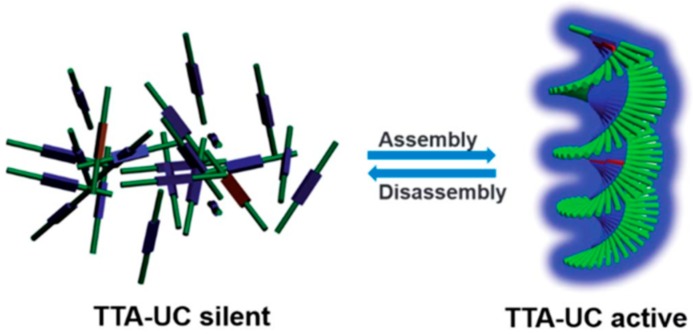
Schematic illustration of the self-assembly-induced TTA-UC. Acceptor molecules spontaneously co-assemble with donor molecules to enable reversible thermal switching of TTA-UC. Adopted with permission from [38]. Copyright 2017 Royal Society of Chemistry.

**Figure 6 gels-05-00018-f006:**
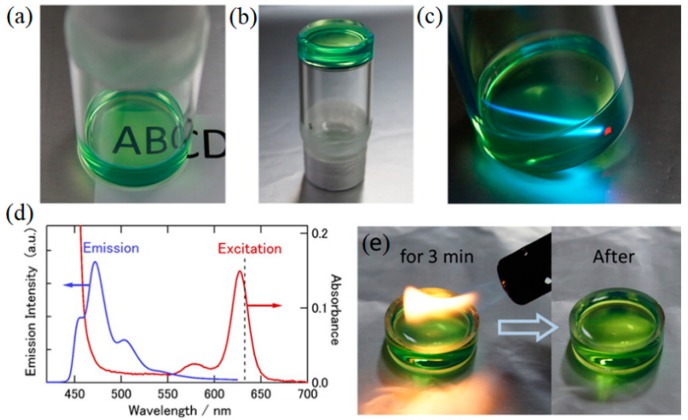
Ionogel photon upconverter prepared using a gelator concentration of 7 g/L. These panels show the (**a**) optical transparency, (**b**) mechanical stability upon inversion, (**c**) upconversion of incident red light (633 nm, ca. 10 mW) to blue light, (**d**) emission and optical absorption spectra, and (**e**) demonstration of the nonflammability by direct exposure to a flame for 3 min. Adopted with permission from [41]. Copyright 2016 American Chemical Society.

**Figure 7 gels-05-00018-f007:**
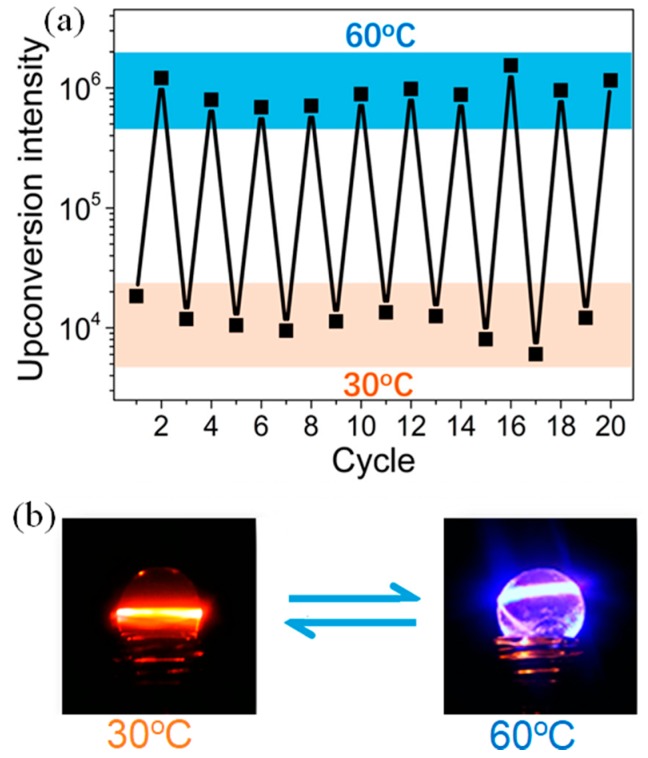
(**a**) Upconversion intensity of the PAAS hydrogel containing PdTPP-DPA-Toluene-Tween-80 microemulsion in the heating-cooling cycles (λex = 532 nm). (**b**) Digital photo of the emission of UC hydrogel at 30 °C (orange-red) and 60 °C (blue). Adopted with permission from [42]. Copyright 2017 American Chemical Society.

**Figure 8 gels-05-00018-f008:**
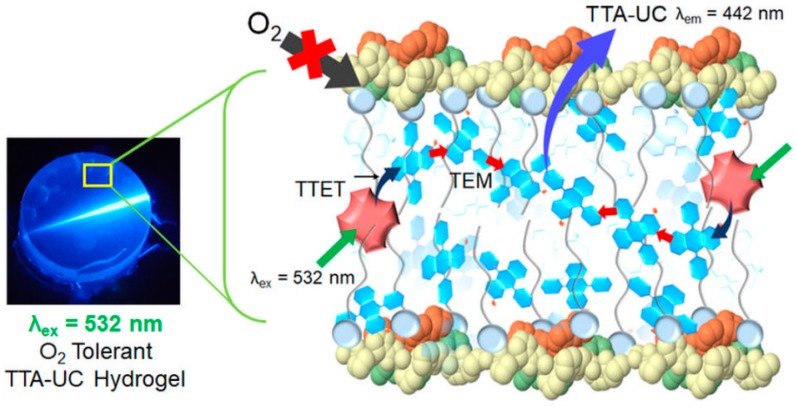
A picture and schematic representation of photon-upconverting hydrogel. Donor (PtOEP) and acceptor (DPAS) molecules are accumulated in the non-polar domains of gelatin-TX100 hydrogel. The donor molecules are excited by long-wavelength light, followed by a sequence of TTET from the donor to the surrounding acceptor, TEM and TTA among the acceptor molecules, and delayed fluorescence from the upconverted singlet state of the acceptor. Adopted with permission from [43]. Copyright 2018 American Chemical Society.

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
