# Peer review of "Recent Progress in Photon Upconverting Gels"

_gels, 2019, doi:10.3390/gels5010018_

Reviewer 1 Report

This manuscript gives a thorough summary on the upconversion gels. The contents are helpful for the readers to have a complete view of the development in the area. I recommend acceptance of the manuscript upon minor revisions.

Line 15, 'organizing' should be changed to 'organized'; 

Line 119, Figure 3c, was a filter used for taking the photo of the upconversion? This issue should be clarified;

Line 163, the full name of the ionic liquid should be given;

A few related review article can be added as references, such as RSC Adv., 2012, 2, 1712–1728; Chem. Eur. J. 2011, 17, 9560 – 9564.

Author Response

Comment: Line 15, 'organizing' should be changed to 'organized'; 

Answer: The word is corrected in the revised MS

Comment: Line 119, Figure 3c, was a filter used for taking the photo of the upconversion? This issue should be clarified;

Answer: No filter was used to take the photograph; this line has been added in the Figure 3C of revised MS. 

Comment: Line 163, the full name of the ionic liquid should be given;

Answer: The full name of IL is already provided in the MS. CDBA6NTf2 is a polymeric gelator and we removed this abbreviation and described it just as “a polymeric salt gelator”, to avoid confusion.

Comment: A few related review article can be added as references, such as RSC Adv., 2012, 2, 1712–1728; Chem. Eur. J. 2011, 17, 9560 – 9564.

Answer: According to the suggestion, these references have been added in the revised MS as ref. 17 and ref. 11, respectively. Accordingly, the reference numbering has been changed.

Reviewer 2 Report

The manuscript presents a review of studies of triplet-triplet annihilation-based photon upconversion in gels. With respect to their air-stability, the molecule diffusion mechanism and the energy migration mechanism in the hydrogen-bonding networks in some types gels containing various solvents such as organic solvents (organogels), ionic liquids (ionogels), and water (hydrogels) are reported in this review. Such systems are valuable to study because they exhibit excellent triplet-sensitized photon upconversion to efficiently absorb light and transfer its energy without oxygen quenching. The studies of their photon upconversion were well reviewed and summarized in this manuscript. The reviewer appreciates the explanation of the present studies as photoscience and technology. However, there may be too many studies of the authors themselves. The authors should provide more studies of the others. Therefore, this manuscript will be acceptable for publication after appropriate revision.

Author Response

Comment:  However, there may be too many studies of the authors themselves. The authors should provide more studies of the others. Therefore, this manuscript will be acceptable for publication after appropriate revision.

Answer: We appreciate the positive response of the reviewer about this manuscript. Regarding reviewers concern of too many reports of authors, it is to be mentioned that we have included all of the reports published in the gel-related upconversion field. Although our research group has published maximum work in this filed, four Figures (Figure 1, Figure 2, Figure 4, Figure 6) out of eight Figures are taken from the work of the other groups, in consideration of the reviewer’s point. Also, we newly added references 11 and 17 according to reviewer 1's suggestion.